# COVID-19 Vaccines: Current and Future Perspectives

**DOI:** 10.3390/vaccines10040608

**Published:** 2022-04-13

**Authors:** Luca Soraci, Fabrizia Lattanzio, Giulia Soraci, Maria Elsa Gambuzza, Claudio Pulvirenti, Annalisa Cozza, Andrea Corsonello, Filippo Luciani, Giovanni Rezza

**Affiliations:** 1Unit of Geriatric Medicine, Italian National Research Center on Aging (IRCCS INRCA), 87100 Cosenza, Italy; l.soraci@inrca.it (L.S.); a.corsonello@inrca.it (A.C.); 2Scientific Direction, Italian National Research Center on Aging (IRCCS INRCA), 60121 Ancona, Italy; f.lattanzio@inrca.it; 3Department of Obstetrics and Gynecology, University of Ferrara, 44121 Ferrara, Italy; giuliasoraci@gmail.com; 4Territorial Office of Messina, Italian Ministry of Health, 98122 Messina, Italy; 5USMAF Sicily, Italian Ministry of Health, 90133 Palermo, Italy; c.pulvirenti@sanita.it; 6Laboratory of Pharmacoepidemiology and Biostatistics, Italian National Research Center on Aging (IRCCS INRCA), 87100 Cosenza, Italy; a.cozza@inrca.it; 7Infectious Diseases Unit of Annunziata Hospital, 87100 Cosenza, Italy; filippoluciani@gmail.com; 8Health Prevention Directorate, Italian Ministry of Health, 00144 Rome, Italy; g.rezza@sanita.it

**Keywords:** COVID-19, recombinant vaccines, pandemic, S protein, innovative approaches

## Abstract

Currently available vaccines against severe acute respiratory syndrome coronavirus-2 (SARS-CoV-2) are highly effective but not able to keep the coronavirus disease 2019 (COVID-19) pandemic completely under control. Alternative R&D strategies are required to induce a long-lasting immunological response and to reduce adverse events as well as to favor rapid development and large-scale production. Several technological platforms have been used to develop COVID-19 vaccines, including inactivated viruses, recombinant proteins, DNA- and RNA-based vaccines, virus-vectored vaccines, and virus-like particles. In general, mRNA vaccines, protein-based vaccines, and vectored vaccines have shown a high level of protection against COVID-19. However, the mutation-prone nature of the spike (S) protein affects long-lasting vaccine protection and its effectiveness, and vaccinated people can become infected with new variants, also showing high virus levels. In addition, adverse effects may occur, some of them related to the interaction of the S protein with the angiotensin-converting enzyme 2 (ACE-2). Thus, there are some concerns that need to be addressed and challenges regarding logistic problems, such as strict storage at low temperatures for some vaccines. In this review, we discuss the limits of vaccines developed against COVID-19 and possible innovative approaches.

## 1. Introduction

High population density, increased contact with animal reservoirs, rapid transport, and massive population movements represent the main determinants of the global spreading of emerging pathogens with pandemic potential [1,2]. In particular, respiratory viruses are able to spread across wide geographical areas in a short period of time, causing high levels of morbidity and mortality. Among airborne pathogens, coronaviruses (CoV) have demonstrated the ability to cause threatening pandemic events [1,2].

Despite the recent emergence of several zoonotic pathogens highlighting the need for global preparedness and the rapid development of vaccines against previously unknown pathogens [3], at the beginning of the SARS-CoV-2 pandemic, there were still no vaccines against human coronaviruses; furthermore, those which have been successfully developed in a few months’ time do not appear to be capable of ensuring universal, long-lasting protection [4,5].

In particular, the emergence of SARS-CoV-2 variants with decreased susceptibility to the neutralizing antibody responses induced by currently available COVID-19 vaccines raises the possibility of breakthrough infections [6]. In addition, convalescent plasma from previously infected individuals does not reduce the risk of hospitalization due to COVID-19 [6]. Thus, alternative or complementary approaches need to be considered in order to develop vaccines able to induce a lasting immunological response and to favor the rapid development and deployment of a high volume of vaccines for pandemic response.

In this review, we discuss the advantages and the limits of currently available vaccines against COVID-19 and the main innovative approaches that might be used to tackle these bottlenecks.

## 2. Epidemiology

The SARS-CoV-2 coronavirus belongs to a large family of enveloped, positive-sense single-stranded RNA viruses capable of infecting many species of birds and mammals, including humans [7].

There are hundreds of coronaviruses, most of which circulate among pigs, camels, bats, snakes, pangolins, and cats. Most human coronaviruses cause nothing more than a common cold [7]. Before the SARS-CoV-2 pandemic, two other zoonotic coronaviruses have undergone species passage, causing several outbreaks: SARS-CoV, which emerged at the end of 2002 in China and was identified in Hong Kong in 2003 [8,9,10], and the Middle East Respiratory Syndrome coronavirus (MERS-CoV), which was identified on the Arabian peninsula in 2012 [11].

The agent causing the current pandemic, SARS-CoV-2, emerged in late 2019 in China and quickly spread throughout the world [12]. As of 28 January 2022, a total of 364,191,494 cases of COVID-19 had been confirmed, including 5,631,457 deaths [13].

The original strain of Wuhan SARS-CoV-2, the cause of COVID-19, probably originated from bats and can be transmitted by asymptomatic, presymptomatic, and symptomatic individuals through close contact via exposure to infected droplets and, to a lesser extent, aerosols [14,15]. This first strain was followed by multiple variants resulting from genetic recombination within infected cells; among them, the Delta variant, formerly called B.1.617.2, and the Omicron variant, formerly called B.1.1.529, recently spread across the world as a result of their increased transmissibility and higher rates of presymptomatic and asymptomatic transmission [16,17].

Human infection due to SARS-CoV-2 can be asymptomatic or cause symptoms that range in severity from mild common colds to critical respiratory illnesses such as acute respiratory distress syndrome and pneumonia [18]; several nonrespiratory symptoms, such as chest pain, abdominal pain, diarrhea, vomiting, cardiac arrhythmias, myalgias, arthralgias, general malaise, headache, and irritability, often complicate the clinical picture of COVID-19 patients [19]. Severe and acute symptoms, often associated with fatal outcomes, appear more frequently in older individuals with comorbidities [18,20,21] and frailty [22,23].

## 3. Genome Structure, Pathogenesis, and Viral Receptor Analysis

SARS-CoV-2, belonging to the *Coronaviridae* family and the *Betacoronavirus* genus [24], is characterized by a polycistronic RNA genome of 27.9 kb containing 14 open reading frames (ORFs), most of which are preceded by a transcriptional sequence. The genome encodes 16 non-structural proteins (NSPs) and 4 main structural proteins: spike (S), envelope (E), core membrane (M), and nucleocapsid (N), and a multitude of accessory proteins are interspersed between the structural genes [12,25,26,27]. The viral envelope is characterized by the E, M, and S proteins, the accessory proteins 3a, 3b, 6, 7a, 7b, 8a, 8b, and 9b, and the N multidomain RNA-binding proteins, that play a critical role in many aspects of the viral life cycle [28]. ORFs 1a and 1b, also known as R/T genes, encode large NSPs, which play roles in host-cell environment modulation and immunological escape [29,30]. The E protein is involved in both the envelope formation and the virus assembly and budding. Although poorly immunogenic [31], it plays an important role in immune responses as it is sensed by specific innate immune receptors, such as Toll-like receptor (TLR) 2, and induces inflammatory cytokine expression [25,32]. Conversely, the M protein possesses high immunogenicity and is capable of strongly inducing T- and B-cell responses with effective neutralizing antibody production [27]; additionally, with its cytoplasmic tail, it contributes to stabilization of the N protein [27,32,33]. The N protein is mainly involved in viral genome packaging and self-assembly and participates in cellular pathway interference mechanisms, thus representing a key regulatory component of the virus [34]. The S protein, a trimeric fusion type I glycoprotein, is mainly involved in the formation of peplomers on the virion surface. This protein contains 2 subunits, S1 and S2. The S1 subunit contains the three receptor-binding domains (RBDs) that specifically bind to the cell surface host receptor, the human angiotensin-converting enzyme (ACE) 2 [34,35]; the S2 mediates the fusion of the viral and host-cell membranes [36,37,38,39]. Since RBDs can induce the production of neutralizing antibodies, the S protein is considered a protective antigen [39,40,41]. In order to be activated and allow the entry of SARS-CoV-2 into host cells, the S protein cleavage site, located at the junction between the S1 and S2 subunits, must be cleaved by specific host enzymes, including transmembrane protease serine protease 2, cathepsin L, and furin [42]; these enzymes are co-expressed with ACE2 in nasal epithelial cells, lungs, and bronchial branches, which explains some of the tissue tropism of SARS-CoV-2 [43,44]. After cleavage, the S protein loses its stability and undergoes a conformational adaption required for binding to the human ACE2 receptor.

Both ACE2 and ACE are membrane-anchored enzymes with wide tissue distribution, and can be shed from the cell surface in soluble forms that retain enzyme activity [39,45,46,47]. ACE2, which operates as an ACE counterpart, is an ectopeptidase that is expressed in 72 tissues, including the brain, heart, lungs and blood vessels [48]. It represents the ligand for RBDs of the SARS-CoV-2 S protein [37,49].

The structural proteins of SARS-CoV-2 can undergo mutations which create more complexities in pathogenesis and for available COVID-19 therapeutic strategies [50]. In addition to stabilizing mutations, there are some combinations of destabilizing mutations, which may increase the risk for interspecies transmission and immunological escape [50,51]. A comparative genomics analysis identified many destabilizing and stabilizing mutations occurring in the structural proteins of SARS-CoV-2, some of them having a great impact on the structure of the virus and its pathogenesis [21,52]. The main immunogenic characteristics and mutations of SARS-CoV-2 are represented in Figure 1.

The possible routes of SARS-CoV-2 transmission among humans include activities such as speaking, breathing, coughing, sneezing, and singing [52]. Following binding to the nasal and respiratory epithelial cell receptors, SARS-CoV-2 propagates and migrates through the respiratory tract and enters the alveolar epithelial cells in the lungs. The rapid replication of the virus in the lungs may trigger a strong immune response, termed the “cytokine storm syndrome”, that causes the acute respiratory distress syndrome and respiratory failure that are considered the main causes of death in patients with COVID-19 [53]; in this regard, older patients or individuals with pre-existing comorbidities such as hypertension, cardiovascular disease, chronic kidney disease, cerebrovascular disease, cancer, diabetes, and obesity, as well as frail older subjects, were shown to have more unfavorable outcomes [21,54].

## 4. Immune Response to SARS-CoV-2

Relevant changes in both innate and adaptive immunity have been reported in COVID-19 patients.

SARS-CoV-2 infects the host respiratory tract through the oral and/or nasal cavities, where the viral mRNA is recognized by the host innate immune receptors, including endogenous TLRs [55,56,57], nucleotide-binding domain and leucine-rich repeat containing receptors [58], and nucleotide-binding oligomerization domain (NOD)-like receptors [58,59]; this interaction triggers the production of proinflammatory cytokines and the induction of an antiviral state in the local tissue environment; proinflammatory cytokines, in turn, stimulate immune effector cell recruitment, and T-cell priming, thus leading to the induction of the adaptive immune response [60,61,62]; this consists of the production of specific antibodies and the activation of CD4^+^ and CD8^+^ T cells, which can each play protective roles in viral infection control [63,64,65,66]. Dysregulation of this initially protective response can cause detrimental systemic hyperinflammation and contribute to tissue damage. In addition, inflammatory cell death causes a further release of cytokines and chemokines, which further contribute to the severe “cytokine storm” that results in acute multi-organ injury [52,60].

However, balanced activation of the innate and adaptive immune systems is important to control the SARS-CoV-2 infection. Indeed, in patients with impaired innate immunity, the virus replicates widely in the upper respiratory tract and the lungs and fails to prime an efficient adaptive immune response, leading to conditions that cause severe lung disease [63,64,65,66,67].

Humoral responses are also key components of adaptive immunity against viral infection [66]. Both immunoglobulins (Ig) alpha (α) and gamma (γ) from COVID-19 patients were shown to mediate viral neutralization and to play a crucial role in immunity at the different stages of infection and in specific anatomical locations [66,68,69,70].

Among the Igγ, S antigen–specific IgG and IgA elicited by viral infection were shown to mediate viral neutralization [70,71]; IgA seems to dominate the early neutralizing response to SARS-CoV-2 [71], which is highly relevant to clinical prognosis [72]. Generally, in a viral infection, the adaptive immune response takes time to develop, and many cells are already infected by the time an antibody response develops [72]. In any case, humoral and cellular immunity cooperate to mount an effective response against the virus infection but with different locations and timing: antibodies are capable of stopping viruses outside of cells, whereas T cells stop viruses inside of cells [73,74]. Altogether, neutralizing antibodies appear detectable within 7 to 15 days of disease onset, and the levels increase until days 14–22 [69,70]; T-cell immunity develops over a period of at least 10–20 days postsymptom onset, and the presence of T cells, together with neutralizing antibodies, is associated with a successful resolution of average cases of COVID-19 [71]. Indeed, a marked decline in the levels of circulating CD4^+^, CD8^+^, B, and natural killer cells appears to be correlated with disease severity and death in COVID-19 patients [61,65,75,76]. This also likely results in delayed neutralizing antibody responses, which are generally T cell–dependent [72]. Furthermore, defective T- and B-cell responses appear to be relevant in the increasing vulnerability to COVID-19 among older individuals, especially in the presence of frailty [71,75,77].

However, although most studies concerning the human adaptive immunity induced by SARS-CoV-2 infection and/or vaccination have been performed using blood samples, local immune responses were shown to be very important in controlling the virus infection. Indeed, since immunological cells and antibodies in the blood do not necessarily reflect what is present in an infected tissue, it could be very important to also understand the relationship between the systemic immune response and the local response in affected tissues [72,78].

## 5. SARS-CoV-2 Antigenic Targets

The selection of protective and immunogenic epitopes represents the most critical step in the vaccine development process. The S and N proteins of coronaviruses are known to be the main targets of the humoral immune response. More specifically, the major SARS-CoV-2 antigenic target of neutralizing human IgM, IgG, and IgA is represented by the RBD of the S protein, followed by the N protein [72,78].

The antibodies directed against the RBD appear to inhibit virus–receptor binding and, consequently, virus entry [79]. Since the full-length S protein represents the best target antigen for vaccine development, mutations have been introduced to block the cleavage of S into the S1 and S2 subunits in order to obtain homogeneous S protein trimers in the prefusion conformation [79]. This construct is the basis for several SARS-CoV-2 vaccine candidates being delivered by adenoviral (Ad) vectors displayed on nanoparticles or encoded by mRNA [79].

The N protein was shown to be highly immunogenic, being capable of inducing strong immune responses in COVID-19 patients; despite its partial homology with the N proteins of other coronaviruses suggesting potential cross-reactivity [79], recent evidence showed that its C-terminus region is specific to SARS-CoV-2 and may represent a protective antigen [80]. In fact, levels of antibodies directed against the N protein increase with the severity and duration of symptoms, and IgG antibodies remain stable for at least 3 months [80].

The reverse vaccinology approach of the in silico study aimed to identify potential B- and T-cell multiepitopes, allowing selection of a further list of promising candidates for vaccine development [79,81]. In addition to S and N antigens, M glycoprotein was reported to exhibit both pathogenic and immunogenic properties [82], whereas NSP3, NSP4, and NSP6 were found both to play a crucial role in the viral adhesion and host invasion and to exhibit high protective immunogenicity [83]. NSP3 was also found to be involved in regulating the host innate immune response by inhibiting the NF-kB signaling and IFN induction pathways [84], whereas NSP6 inhibits the transfer of viral components to lysosomes [84]. In addition, the ORF3 proteins, containing many immunogenic regions, are involved in inflammation and viral escape through the activation of NF-κB and the NOD-like receptor family pyrin domain-containing (NLRP) 3 inflammasome pathways [84].

## 6. Viral Vaccine Platforms

Among the most cost-effective strategies for preventing viral infections, vaccination represents the best tool for helping the immune system to activate protective responses [85,86]. Indeed, vaccination can contribute to preventing or controlling the spread of contagious viral diseases by activating the host immune system to induce long-term immune memory [66], as shown in Figure 2.

In the case of global public health emergencies, governmental vaccine design can benefit from a range of platform technologies, including conventional vaccines such as inactivated and live-attenuated vaccines, the innovative new class of DNA- and RNA-based vaccines [61,87], and promising protein-based vaccines [88,89].

Compared with conventional vaccines, molecular-based platforms may offer a more versatile tool against new emergent viruses, allowing fast, low-cost, and scalable vaccine manufacturing. Essentially, these platforms rely on the use of a system to deliver and present a new antigen (or a synthetic gene) to rapidly target an emergent pathogen. Currently, there are four different platforms used to develop viral vaccines: whole virus, nucleic acid-based, viral vectors, and protein and virus-like particles (VLPs). The choice of platform takes into account many factors, including the way the immune system responds to the specific viral infection, vaccination strategies and policies, and the best technology or approach to create the specific vaccine.

A general concern for the development of vaccines is represented by antibody-dependent enhancement (ADE) because the mechanisms that underlie antibody protection against any pathogen have a theoretical potential to amplify the infection or trigger harmful immunopathology [90]. In viral ADE, the binding of virions to non-neutralizing or subneutralizing antibodies can always promote a more efficient viral uptake into the target cell in Fcγ-receptor- (FcγR)- [87] or complement-dependent-mediated mechanisms [91], leading then to enhanced viral replication and pathogenicity. These data support the hypothesis that purified antigens represent a better strategy to avoid or reduce the risk of detrimental effects related to ADE.

### 6.1. Whole Virus Vaccines

There are two types of whole virus vaccines: inactivated and live-attenuated viral vaccines. In inactivated viral vaccines, the virus is rendered noninfectious by using chemicals such as formaldehyde, radiation treatment, or heat [89]; the viral genetic material does not replicate but is able to trigger the host immune response. In contrast, live-attenuated viral vaccines consist of weakened or attenuated viruses, which are able to replicate within cells and to activate innate immune responses [89].

Despite both vaccines containing the whole or part of the virus, the quality of the immune responses they are able to trigger is different; indeed, inactivated vaccines can only provide a humoral immune response and need several doses over time in order to get protective immunity. In contrast, live-attenuated vaccines are more effective and induce a strong immune response, both cellular and humoral, with just one or two doses. However, in some cases, they can cause adverse events in people with weakened immune systems, in those with long-term health problems, or in people who received an organ transplant. In addition, they need to be kept cool, and therefore a significant logistical mode of distribution is needed in order to provide a vaccine with high efficacy and without logistical challenges during distribution [61].

### 6.2. Nucleic Acid-Based Technologies: DNA and mRNA Vaccines

Nucleic acid-based technologies employ either plasmid DNA or mRNA coding for the antigen(s) of interest and are directly injected into the recipient, whether human or animal, with the antigen subsequently produced in situ by cells of the vaccinee [92]. These vaccines are extremely versatile since they allow the direction of the in vivo assembly of encoded vaccine antigens, thus bypassing the requirement for in vitro antigen expression, assembly, and purification. In addition, the production of antigens inside the target cells allows the obtaining of post-translational modifications with a high degree of faithfulness [86,92]. Despite DNA encoding, viral antigens suffer from low immunogenicity [93], and nanotechnology-based platforms also provide innovative methods to improve specific delivery and sustainable release of antigens and to reduce off-target side effects [94]. In particular, the introduction of adjuvants combined with immunoregulatory agents contributes to the control of improper immune stimulation and to the loss of bioactivity of immunostimulatory agents during circulation [94,95], allowing the obtaining of DNA and mRNA vaccines capable of inducing both humoral and cellular immune responses.

#### 6.2.1. DNA-Based Vaccines

DNA vaccines consist of plasmid DNA-encoding antigens that are expressed in host cells as native antigens with species-specific post-translational modifications [96]. By using administration devices, such as a gene gun or electroporation, DNA vaccines must be introduced into the host cell nucleus; the vaccine DNA can then be transcribed into mRNA that in turn migrates into the cytoplasm, where it is translated into antigenic proteins. Antigen-presenting cells process these proteins and present them to lymphocytes that have become capable of killing pathogens and infected cells that express these target proteins.

Despite DNA vaccines being capable of eliciting both humoral and cellular immunity in theory [94,95,97], the main challenge that limits clinical application is the blocking of their delivery via targeted immune cells which obstruct the stimulation of robust antigen-specific immune responses in humans [94,95,97]. However, the immunological response can be enhanced by several techniques, including complexing the DNA with lipids or polymers in order to decrease the size for easy passage through the cell membrane [94,95,97].

The use of DNA vaccines carries other disadvantages. The introduction of foreign genetic information into the nucleus of the transfected host cells is associated with future oncogenic potential, which is additionally enhanced by the long-term persistence of DNA plasmids upon injection [94]. Another problem in vaccinated organisms can be the potential expression of specific antibiotic-resistance markers originating from antibiotic resistance genes used for the production of plasmid DNA, although these markers have been replaced in next-generation DNA vaccines. Finally, the expression of cytokines or the costimulatory molecules always used to enhance DNA immunogenicity might induce adverse effects, including inflammatory responses, autoimmunity, or immune suppression [98]. Furthermore, despite advancements in adjuvant- and nanomaterial-based delivery systems, DNA vaccines fail to generate potent immunogenic responses [94,95] and require repeated injections of large amounts of plasmid DNA [94,97,99].

#### 6.2.2. RNA-Based Vaccines

Unlike DNA vaccines, RNA vaccines are translated upon crossing the plasma or endosomal membranes, without the need for entering the nucleus. Two major types of mRNA vaccines were developed against infectious pathogens: conventional nonreplicating mRNA vaccines and self-amplifying vaccines (or viral replicons) [86,100].

Nonreplicating (or nonamplifying) RNA vaccines are the simplest type and consist of small mRNA molecules encoding only the antigen of interest and containing 5′ and 3′ untranslated regions [100,101]. Among the main advantages of using nonreplicating mRNA vaccines are the simplicity of the construct, the small molecular size, and the absence of any additional encoded proteins that could induce unintended immune responses [100].

In contrast, self-replicating RNA vaccines are large mRNA molecules derived from specific viral genomes are commonly based on an *alphavirus* or *flavivirus* genome backbone that is defective in at least one structural protein gene but contains genes encoding the viral antigen as well as the viral replicase enzyme machinery that allows their self-amplification [101].

Consequently, compared with conventional nonreplicating mRNA vaccines, self-amplifying RNA vaccines allow intracellular antigen-encoding RNA amplification and higher antigen expression levels [101]. However, the bigger size of these vaccines makes manufacturing processes more difficult, with lower yields and an increased occurrence of abortive constructs. In order to act as a vaccine, exogenous mRNA has to enter the cytoplasm by crossing the plasma or endosomal lipid membrane, and once it is delivered into the cytoplasm of the host cell, it can be translated by the host translational machinery, producing the corresponding post-translationally modified antigen, thus mimicking an in vivo viral infection. Several studies showed that mRNA vaccines can efficiently trigger the activation of the innate immune system, also inducing strong adaptive immune responses [102,103,104]. The intrinsic innate immune stimulation properties of mRNA can be further increased by implementing different mRNA formats and formulations. A more commonly used strategy to increase expression and immunogenicity is the delivery of mRNA in complex with additional components, such as the protamine-complexed mRNA format that has been shown to provide both strong antigen expression and efficient immunostimulation [104,105,106].

The efficacy of mRNA vaccines can be further improved by adding safe complexing agents such as lipid- and polymer-based nanoparticles, which enhance uptake by cells, improve delivery to the translational machinery in the cytoplasm, and are also well tolerated [107,108,109]. The most commonly used components for lipid-based nanoparticle formulations are represented by ionizable amino lipids, phospholipids, cholesterol, and lipid-anchored polyethylene glycol. Since mRNA vaccines are fully synthetic, they do not need a biological phase, thus being able to gain access to clinical trials in a very short time [102].

However, despite naked mRNA being able to induce immune responses, RNA vaccines containing mRNA alone are not applicable for broad use as prophylactic vaccines since the ubiquitous presence of extracellular ribonucleases catalytically hydrolyzes RNA [110]. Altogether, since the production process of mRNA is based on in vitro systems and does not require amplification in bacteria or cell cultures, the manufacturing of mRNA vaccines is a comparably shorter and simpler process to monitor. Furthermore, given that mRNA vaccines do not enter the nucleus, they avoid the potential risk of genomic integration posed by DNA-based vaccines. Most mRNA vaccines can be administered by different routes using conventional needle-based injections and, unlike DNA vaccines, without requiring any additional administration devices. Therefore, mRNA vaccines offer a flexible one-for-all, large-scale, rapid and cost-effective manufacturing process with fast turnaround times. These characteristics are very important to counteract pandemic threats that require a rapid response platform capable of producing protective vaccines in a short time frame in order to protect at-risk populations and to have an early impact on the progression of an outbreak.

### 6.3. Recombinant Viral Vector-Based Vaccines

Replicating recombinant vector vaccines consists of a fully competent viral vector backbone engineered to express an antigen from a foreign transgene [110]. In the viral vectors, the synthetic gene encoding the antigen(s) is inserted into one of the specific viruses that usually have been engineered so that they cannot replicate in the human host. The virus is then grown in culture and used to deliver the synthetic gene during vaccination.

Recombinant viral vaccine vectors have been shown to induce robust and long-lasting immune responses, thus providing protective efficacy against a variety of infectious disease threats to humans and animals, including those with the zoonotic potential to cause global pandemics. Several types of human and animal viruses, including chimpanzee Ad, human Ad 5 and 26, the measles virus, vaccinia Ankara, the vesicular stomatitis virus, and the cytomegalovirus have been modified for use as vector-based vaccines [111].

In particular, recombinant Ad vector-based vaccines have been shown to be very attractive as they induce both innate and adaptive immune responses in mammalian hosts.

Currently, Ad vector-based vaccines are being used to counteract numerous infectious diseases, ranging from malaria to HIV-1 [112]. Additionally, these vaccines can stably express inserts of up to 8 kb, supporting the expression of most target antigens, thus acting as multivalent or multipathogen vaccines [113]. Furthermore, they support high viral yields at relatively low production costs. Ad vector-based vaccines are also able to induce potent antibody as well as T-cell responses, with variations in the immune response depending on the serotype employed [114].

However, in some cases involving the use of human Ads as vectors, the widespread pre-existing immunity in the human population can inactivate the viral vector, thus inhibiting transgene expression [115]. This issue has been met by developing Ad vectors of nonhuman origin or, as an alternative approach, by selecting rare serotypes with low prevalence in humans [113,116].

Administration of viral vector-based vaccines can take place through different routes of administration, including intramuscular, intranasal, intradermal, and oral, depending of the target pathogen [117,118,119]. Viral vectors induce infection-like stimuli within target cells, thereby enhancing potent immune responses without the need to add adjuvants [120,121].

The valid strategies to achieve replication incompetency or attenuation in order to obtain a good safety profile, together with high yield production processes, support the use of these viral vectors for pandemic settings.

However, since viral vectors are genetically modified organisms, they represent a potential risk to human health and environment. Moreover, their use as attenuated vector-based vaccines could lead to high or persistent replication or to their integration into the host genome. In addition, since the production of these vaccines involves multiple steps and components, including cells and substrates, it requires extensive testing to exclude the presence of contaminants [122]. These factors, together with the screening procedures to evaluate potential pre-existing immunity, make the production of viral vector-based vaccines a highly complex and cost-intensive process.

### 6.4. Recombinant Protein-Based Vaccines

Many recombinant proteins, realized through genetic engineering techniques, have historically been used as vaccine candidates [123]. Protein subunit vaccines are made of one or more purified antigens from the viruses or bacteria of interest that are able to trigger the immune system and raise a protective immune response. The first subunit vaccine was approved in 1986 and directed against the hepatitis B virus (HBV) by including a recombinant viral surface antigen (HBsAg) able to self-assemble using *S. cerevisiae* as an expression system. However, recombinant protein vaccines are, per se, poorly immunogenic and require association with a specific adjuvant(s) to increase the induction of long-lasting protective immune responses [124]. Thanks to the absence of infectious components, these vaccines are considered safer compared with those derived from live viruses or recombinant genetic material [125].

### 6.5. Virus-like Particles (VLPs)

VLPs are made of multimeric viral proteins which are similar to native viral particles but are not infectious and do not contain genetic material. Multimers of proteins are responsible for the formation of chimeric VLPs (cVLPs) that stimulate protective immune responses via activation of T and B cells [126]. Because of poor immunogenicity, VLPs need to be complexed with adjuvant molecules. After being recognized as naïve molecules by the host immune system, they are capable of boosting immune responses. These types of vaccines have the advantage of being relatively safe, adaptable, and less expensive compared with other platforms [127]. Depending on their structural composition, VLPs can be classified as nonenveloped (neVLPs) or enveloped (eVLPs) candidates. Both kinds can then be composed of native viral antigens only (homologous VLPs) or can contain a mixture of proteins from different sources to increase overall immunogenicity (heterologous VLPs) [127].

## 7. Vaccinology in the COVID-19 Pandemic Era

Since the emergence of the pandemic, several vaccines have been developed and received EUA against SARS-CoV-2 [128,129]. As of 8 February 2021, there were 180 vaccine candidates and 33 approved vaccines for SARS-CoV-2 infection [129].

Different approaches have been used in parallel to make COVID-19 vaccines, including the use of nucleic acid-based vectors [130,131], whole virus (live-attenuated and inactivated), viral vectors (replicating and nonreplicating) [132,133,134], adjuvant recombinant protein nanoparticles [135], and virus-like particles (VLPs). Among the protective antigens of SARS-CoV-2, the attention has mainly focused on the native S protein, which is able to induce potent neutralizing antibodies, even if its presentation to the immune system differs substantially between the different categories of vaccines [136]. However, new evidence is being raised about potential roles for other, more conserved non-spike viral antigens, such as nucleocapsid (N) proteins, which might represent an innovation in the fight against emerging SARS-CoV-2 variants and a source for universal vaccines providing long-lasting immunity [137].

Many vaccines approved for emergency use against COVID-19 contain biomaterial and nanoparticles that facilitate the delivery and action of the vaccine within host cells [138]; indeed, the mRNA vaccines developed by pharmaceutical companies consist of lipid nanoparticles containing the nucleic acid sequence encoding the S protein, or in recombinant Ad vectors, containing the S antigen–encoding sequence in the Ad DNA [130,131,139,140,141,142,143,144,145,146,147]. The main nanoparticles used to enhance the delivery of the vaccine into the host cells are shown in Table 1.

Some nanotechnology-based companies have developed alternative approaches based on the conjugation of the S protein onto the surface of nano-sized VLPs for effective delivery of the antigen protein to the host body [149,151]. Among the adjuvants used to enhance the immune response against the S protein, aluminum-based nanoparticles have been shown to improve the immunogenicity in groups that respond poorly to vaccines [141,144,151]. These adjuvants show a high efficiency in dendritic cell cross-presentation and the subsequent induction of cellular immunity. In addition, the use of TLR ligands in combination with aluminum nanoparticles appears to enhance the cross presentations by dendritic cells [152].

### 7.1. Whole Virus Vaccines

Whole virus vaccines contain attenuated or inactivated forms of SARS-CoV-2 that can stimulate a protective immune response without causing the disease. The immunogenic profile of these two types of vaccines is clearly different. Whereas live-activated vaccines can stimulate both cellular and humoral immune responses, inactivated vaccines can only stimulate humoral response to SARS-CoV-2 [153]. However, use of live-activated forms of the virus raises several problems that limit the use of these platforms, such as postvaccination viral mutations inducing increases in toxicity and adverse reactions secondary to proliferation of the virus in the nasal cavity [154].

Among inactivated vaccines, 10 candidates have been approved for emergency use authorization (EUA), while 21 candidates were still in the evaluation stage as of 8 February 2022 [129]. The approved vaccines are BBIBP-CorV (89 countries), CoronaVac (53 countries), BBV152 (13 countries), KoviVac (3 countries), inactivated Vero Cells (2 countries), Qaz-COVID-in (2 countries), KConecaVac (2 countries), ERUCOV-VAC (Turkey), FAKHRAVAC (MIVAC; Iran), and COVIran Barekat (Iran). On the other hand, a single live-attenuated vaccine is in a phase 1 clinical trial in the UK [129].

### 7.2. DNA- and RNA-Based Vaccines

In DNA- and RNA-based vaccines, the genetic material from SARS-CoV-2 is used to elicit a protective immune response without causing disease. Both vaccine platforms can induce mainly B- and T-cell responses, with different risk profiles; indeed, DNA vaccines carry the risk of the integration of genetic material within the host DNA; RNA vaccines do not carry this risk but need to be stored at lower temperatures compared with DNA vaccines [153]. As of 8 February 2022, there was 1 DNA-based candidate (ZycoV-D) approved for EUA in a single country (India), while 16 candidates were in the clinical evaluation stage [129]. On the other side, three RNA-based candidates have been used as authorized vaccines for emergency use worldwide: BNT162b2 (137 countries), mRNA-1273 (85 countries), and TAK-919 (Japan); furthermore, 32 RNA-based candidates are currently in the clinical evaluation stage [129].

### 7.3. Viral Vector Vaccines

Viral vector vaccines do not contain antigens by themselves but use the translation mechanisms of the host cells to produce them. Those developed against SARS-CoV-2 are made of modified viral vectors that deliver genes encoding the surface spike proteins; there are two main types of viral vector vaccines: replicating viral vector vaccines enter the host cells and replicate to generate whole viral particles which produce the SARS-CoV-2 spike protein; and nonreplicating viral vector vaccines directly generate viral antigens within the host cells [155]. As of 8 February 2022, six nonreplicating viral vectors had been approved for EUA worldwide: AZD1222 (137 countries), Ad26.COV2.S (106 countries), Gam-COVID-Vac (74 countries), Covishield (47 countries), Sputnik Light (26 countries), and Ad5-nCoV (10 countries). Another 28 nonreplicating viral vectors are at the clinical trial stage. On the contrary, only eight replicating viral vectors are at the clinical trial stage [129].

### 7.4. Protein Subunit Vaccines

There are two main types of protein subunit vaccines against SARS-CoV-2: polysaccharide and conjugate vaccines. Polysaccharide vaccines contain polysaccharides of the SARS-CoV-2 cell wall, while conjugate candidates are bound to a polysaccharide chain with a carrier protein to induce a booster effect in the immune system response [156]. Most protein subunit vaccines contain harmless recombinant spike proteins; however, the emergence of variants of concern (VOCs) has stimulated research in order to identify potential new antigenic targets. In this regard, a recent study showed the potential role of intraviral nucleocapsid (N) protein use as a novel vaccine strategy to target multiple SARS-CoV-2 variants and thus lead to universal long-lasting immunity [137]. In fact, in contrast to the spike protein, the N-protein is relatively conserved within a single strain and among different strains across evolutionary stages. Interestingly, anti-N immunoglobulins have been detected in patients with SARS-CoV and SARS-CoV-2 infections [137,157].

In any case, protein subunit vaccines represent a promising solution for vaccines against COVID-19 as they can also be produced by large-scale microbial fermentation in developing countries, do not require storage at cold temperatures, and are safe when used in combination with adjuvants [158,159]. These vaccines represent a valuable source for large-scale vaccination programs in low- and middle-income countries (LMICs).

To date, 13 protein subunit vaccines have been authorized for emergency use against SARS-CoV-2: NVX-CoV2373 (34 countries), CIGB-66 (6 countries), FINLAY-FR-2 (4 countries), COVOVAX (3 countries), RBD Dimer (3 countries), MVC-COV-1901 (2 countries), FINLAY-FR-1A (Cuba), BECOV2A (India), EpiVacCorona-N (Russia), CHO cell (United Arab Emirates), Razi Corv Pars (Iran), and Covax-19 (Iran) [129]. Furthermore, 60 protein subunit vaccines are currently at the clinical trial stage [129].

### 7.5. Virus-like Particles (VLPs)

Several VLP platforms can be used to produce eVLPs and neVLPs to generate potentially long-lasting immune responses against SARS-CoV-2. To improve the self-assembly of VLP multimers, most candidates were developed combining the N protein and the highly immunogenic spike protein. To address issues related to the fast, large-scale production of VLPs, heterologous VLPs are preferred over homologous molecules. As of 8 February 2022, only five vaccine candidates were at the clinical trial stage, with no VLP approved for EUA against SARS-CoV-2 [129].

## 8. Efficacy, Immunogenicity, and Safety of the Current COVID-19 Vaccines

### 8.1. Efficacy and Immunogenicity

Phase III clinical trials first demonstrated a high vaccine effectiveness (VE) for several vaccines against symptomatic COVID-19 from the original Wuhan strain, such as 95% VE for the BNT162b2 mRNA vaccine (Pfizer-BioNTech) [160], 94.1% VE for the mRNA-1273 vaccine (Moderna) [131], 70.4% VE for the ChAdOx1 nCoV-19 vaccine (AZD1222 Oxford-AstraZeneca) [132], and 50.7% VE for the inactivated CoronaVac [161]. VE was evaluated at least 7 days after the second dose for the BNT162b2 mRNA vaccine and 14 days after the second dose for the other vaccine platforms. Results of a recent metanalysis showed a global 97% VE for the prevention of hospitalization and severe disease and a 99.0% VE for the prevention of COVID-19-related mortality [128,162]. The emergence of the Delta and Omicron variants partially decreased the efficacy of two-dose vaccine regimens, which was specifically contrasted via administration of third and booster doses to enhance protection versus symptomatic and severe COVID-19 [162,163,164,165]. Indeed, vaccination strategies represent the best option to fight SARS-CoV-2 infection and improve the prognosis of infected patients. However, the successes of vaccination campaigns are challenged by breakthrough infections, which are mainly caused new variants and the waning of vaccine-induced immunity [166,167].

Due to the mutation-prone nature of the spike protein, the emergence of new variants of concern may increase the risk of immunological escape, thus decreasing the effectiveness of a vaccine [168,169,170,171]. Several studies have shown that there are efficacy differences related both to population or geography and to locally circulating variants [172]. Analyses of data on neutralizing antibodies, which are considered highly predictive of immune protection [172,173,174,175], highlighted that recombinant S-protein vaccines can induce the highest neutralizing antibody levels [176], followed by mRNA vaccines [130,131,176,177,178,179], and inactivated vaccines [51,180,181,182].

A second potential cause of breakthrough infections is the waning of vaccine-induced immune protection over time; several prospective studies have suggested that antibody levels after mRNA vaccines and the Ad26.COV2.S vaccine could progressively decrease after the second dose [183,184,185]. Additionally, the mutation-prone nature of the S protein may affect the long-lasting immunity associated with both protein and mRNA vaccines [182]. In particular, very recent studies evidenced that two doses of the current COVID-19 vaccines are unlikely to protect against infection by the newly emerged SARS-CoV-2 Omicron variant, and a third dose provides only some protection in the immediate term but substantially less than against previous SARS-CoV-2 infections [164]. Another weakness of the current COVID-19 vaccines is represented by their inability to elicit mucosal immunity. Recent studies showing that SARS-CoV-2-specific IgA antibodies provide more efficient neutralization than IgG highlight the potency of IgA, especially in the early stage of the COVID-19 disease [186]. Mucosal immunity appears to play a very important role in COVID-19 since it represents the most important immunoglobulin for fighting the infectious pathogen at the portal of entry into the respiratory system. Consequently, secretory IgA can act as an immune barrier able to neutralize SARS-CoV-2 by preventing it from reaching and binding to the epithelial cells of the nasal or oral cavities [187]. Mucosal immunity also plays an important role in contributing to asymptomatic infections interference in viral persistence and circulation in the environment [187].

Altogether, the immune responses elicited by current COVID-19 vaccines, including both T-cell responses and neutralizing antibody production, could be improved, within certain limits, by optimizing the primary antigen sequence, the doses, the adjuvants, the immunization regimes, the manufacturing processes, etc.

### 8.2. Safety

An ideal vaccine should be able to trigger both the humoral and cellular arms of the immune system, with reduced potential risks of the phenomena of disease enhancement, also known as vaccine-enhanced respiratory disease (ERD) [188].

Vaccine-associated ERD refers to clinical infections affecting individuals exposed to a wild-type pathogen after having received a prior vaccination for the same pathogen. This phenomenon is mainly characterized by the predominant involvement of the lower respiratory tract and induces a worsening of disease through a mechanism called antibody-dependent enhancement (ADE) [187].

This adverse event, which is also observed in nonhuman primate animals immunized with candidate vaccines against severe acute MERS, is mediated by non-neutralizing antibodies or by T-helper-2-cell-skewed response [188]. However, it should be noted that ADE and ERD were observed both in COVID-19 patients and SARS-CoV-2 vaccinated subjects [188,189]. Theoretically, a higher risk of inducing pathologic vaccine-associated ADE and/or ERD is mainly related to inactivated viral vaccines containing antigens capable of inducing non-neutralizing antibodies and/or the S protein in non-neutralizing conformations. Although non-neutralizing antibodies can specifically recognize the virus, they are unable to prevent the infection; conversely, such antibodies could enhance inflammatory responses, inducing immunopathologic injury. For this reason, one way to decrease the risk of ADE includes the induction of high doses of potent neutralizing antibodies rather than lower concentrations of the non-neutralizing antibodies that would be more likely to cause ADE [188].

Another very serious condition that can be induced by immunization with specific Ad COVID-19 vaccines, including chimpanzee or human Ad vectors, is represented by cerebral venous sinus thrombosis with thrombocytopenia; this potentially life-threatening condition can occur within two-to-three weeks of vaccination, with a mechanism similar to autoimmune heparin-induced thrombocytopenia [189]. Despite its uncertain pathogenesis, many patients with concomitant thrombosis and thrombocytopenia have shown positive or strongly positive results in laboratory analyses for antiplatelet factor (F)4-polyanion autoantibodies [188,190,191].

Myocarditis, mainly occurring in young adult and adolescent males, has been recognized as a further potential complication of COVID-19 mRNA vaccination [192].

In order to explain the mechanisms for the development of myocarditis following COVID-19 vaccination, it has been hypothesized that a molecular mimicry between the S protein and self-antigens is capable of triggering pre-existing dysregulated immune pathways in certain individuals. The male predominance in myocarditis could be explained by both the sex hormone differences in immune response and myocarditis and the underdiagnosis of cardiac disease in women [192].

However, both thrombotic events and myocarditis can be also induced by the COVID-19 disease [191,192]. Indeed, several patients diagnosed with COVID-19 developed a prothrombotic state that placed them at a dramatically increased lethal risk. Recent studies showed that S protein can directly induce platelet activation, such as platelet aggregation, through binding to ACE2, which is also expressed by platelets. Such platelet activation can then lead to thrombus formation and inflammatory responses in COVID-19 patients [191,192,193]. Despite being less common, myocardial injury also occurs among COVID-19-affected patients, leading to increased risk of death [192].

Altogether, despite rare cases of cerebral venous sinus thrombosis with thrombocytopenia and myocarditis, the available data appear to indicate a positive benefit–risk profile for COVID-19 vaccination, with a favorable balance for all age and sex groups [194,195].

Another adverse event related both to SARS-CoV-2 infection and S protein mRNA-based vaccination is represented by hypertension [196,197].

In order to explain this reaction, recent studies hypothesized that autoantibodies against ACE2 may develop as anti-idiotypic antibodies directed against anti-S protein antibodies [193].

More specifically, since the S protein binds to the host ACE2 receptor, the anti-S neutralizing antibodies produced are characterized by the variable regions of heavy (VH) and light (VL) chains arranged in a tetrahedral configuration very similar to the ACE2 receptor, thus mimicking the RBD of ACE2. According to the idiotypic network theory, complementary idiotype structures can be found not only on antigens but also on the antibodies to different idiotypes [198,199]. In this case, the first host antibody recognizes the RBD of the S protein, whereas the second antibody is a host anti-idiotypic antibody that can bind both the RBD of the first antibody and the RBD of the host ACE2 protein, inducing its inhibition. Since ACE2 autoantibodies and lower activity of soluble ACE2 have been detected in the plasma of many patients with a history of SARS-CoV-2 infection, it can be hypothesized that the S protein can trigger ACE2-autoantibody production [200]. In light of these studies, it can be supposed that the same process can occur following vaccination inducing S protein expression in the host (Figure 3).

Figure 3 shows a comparative analysis concerning the adverse effects related to ACE2 following SARS-CoV-2 infection and SARS-CoV-2 mRNA vaccination.

In particular, Figure 3A shows the SARS-CoV-2 is able to stimulate innate immune cells through both exogenous and endogenous TLR ligands, thus inducing a strong TLR signaling leading to cytokine storm, which in turn induces the cascade of adverse events mainly consisting of arterial thrombosis, venous thromboembolism, pulmonary edema, and interstitial inflammation. At the same time, the S protein expressed on the surface of infective particles binds to ACE2 is present both as a membrane-anchored enzyme with wide tissue distribution and in soluble form that retains the enzyme activity. In any case, the S protein induces anti-S-protein antibody production that, in turn, induces anti-idiotypic antibodies able to bind to the RBD of both the anti-S-protein antibodies and ACE2, thereby inducing specific inhibition.

In Figure 3B, the effects potentially related to mRNA vaccination are described.

More specifically, the mRNA encoding the S protein encapsulated in liposomal nanoparticles is phagocytized by innate immune cells and stimulates endogenous TLRs. At the same time, mRNA is translated into S protein inside the immune cells, which can then present it in association with HLA class II molecules, constitutively expressed on professional antigen-presenting cells such as dendritic cells and monocytes/macrophages.

The antigen presentation is known to play a critical role in the generation both of the humoral and cellular immune responses with the survival of memory CD4^+^ T cells. Anti-idiotypic antibody production with consequent inhibition of ACE2 activity can also occur in mRNA vaccination, although in this case the phenomena linked to cytokine production appear to be more attenuated, mainly because of a more limited number of stimulatory agents compared with virus replicating particles, and an innate immune stimulation limited to endogenous TLRs. A brief summary of common and rare adverse effects related to COVID-19 vaccines is reported in Table 2.

## 9. Future Perspectives

The main strategy to counteract the COVID-19 pandemic consists in designing vaccines capable of inducing long-lasting immunity and also of protecting against the circulating variants of SARS-CoV-2.

Mutations in virus sequences, leading to mutated clusters in the structural proteins, play a critical role in limiting the working life of a vaccine. Whole genome analysis of SARS-CoV-2 is important to identify the regions in the virus genome where mutations play an important role, and also in order to understand the dynamics of virus evolution and to guide researchers in designing a sustainable vaccine. Immunogenetic virus proteins, detected by bioinformatics approaches and computational analyses can be analyzed by identifying the points of mutation. The biochemical properties of single-point mutations can then be investigated [206]. These strategies may contribute to generating next-generation vaccines, to containing virus mutation sequences, and to comparing the global circulating variants with those previously identified [79,207,208].

Among other immunogenic SARS-CoV-2 antigens, NSPs have been shown to be crucial for virus adhesion and host invasion [82] and might be used as alternative vaccine targets, with the aim to protect the host against the viral infection by inducing anti-NSP antibodies capable of countering viral entry and replication. The use of NSPs, alone or in combination with the S protein, could also contribute to reducing the dose of each antigen, including the S protein, with consequent reductions in the occurrences of adverse events. In this regard, new vaccine candidates containing N antigen instead of mutation-prone spike protein could be used for ensuring long-lasting immune protection against multiple VOCs [137].

Reverse vaccinology represents another promising bioinformatics approach potentially capable of improving vaccine production and vaccination protocols. This widely used strategy to identify potential vaccine candidates consists of the proteomic screening of a pathogen through computational analysis in order to obtain the in silico prediction of protein-vaccine candidates from genome sequences and to perform a comparative molecular similarity analysis between specific virus antigens and human proteins. Reverse vaccinology can also allow for the elimination of virus antigenic epitopes that share sequences similar to the human proteome, thereby reducing the risk of autoimmune adverse effects [208,209]. Indeed, molecular mimicry is one of the main concerns related to current S glycoprotein-based vaccines. In particular, in addition to the effects caused by the binding of the S protein to human ACE2, this molecule was shown to share sequences similar to alveolar lung surfactant proteins. These sequences appear to induce the production of self-reactive antibodies, which may contribute to the pathologic scenario that accompanies SARS-CoV-2 infection, consisting mainly of acute respiratory distress syndrome, lung damage, and inflammation mechanisms [210].

In the context of SARS-CoV-2 vaccine development, peptide-based phage display represents an inexpensive and versatile tool for large-scale immunization. Phage-displayed vaccines are made by expressing multiple copies of an antigen on the surface of immunogenic phage particles, thereby eliciting a powerful and effective immune response [211]. This approach takes advantage of inherent properties of these particles, such as their adjuvant capacity, economical production, and high stability [211]. Phage particles are easy to genetically engineer or modify and produce in large quantities. In addition, they are capable of stimulating both cellular and humoral immunity and are stable under harsh environmental conditions, including pH and temperature. Therefore, phage display-based vaccine approaches could represent versatile platforms for rapid production of COVID-19 vaccines that would be cost-effective, needle-free, and safely stored long-term at room temperature. Combined with bioinformatics approaches to identify effective epitopes, this strategy provides a promising framework for vaccine development [212]. However, one of the main challenges in designing phage display vaccines is assuring that, following the insertion of the different SARS-CoV-2 antigen epitopes into the phage particles, the structures of these peptides is maintained intact and in their original three-dimensional conformations.

New vaccination techniques could be oriented towards increasing the protection of vulnerable segments of the population, including older patients with frailty. Recent evidence showed that, compared with age or comorbidity alone, frailty itself represents a better outcome predictor in older individuals with COVID-19 [213]. Frailty is defined as a geriatric syndrome characterized by higher vulnerability to minor stressors, which exposes affected individuals to a high risk of adverse health outcomes, including disability and death [214]. COVID-19 vaccination in older, frail individuals is still challenging because they may experience both benefits and harm from vaccination. On one hand, frail, older individuals were shown to experience more severe COVID-19 disease courses compared with prefrail or nonfrail individuals [215]; furthermore, defective activation of T cell lymphocytes during COVID-19 may contribute to poor survival among frail individuals. On the other hand, some cardinal features of aging and frailty, such as inflammation and immunosenescence, contribute to worsening immunological response to vaccination [216], making this vulnerable group potentially more susceptible to adverse reactions due to vaccination [217]. To date, studies supporting the efficacy of vaccination in such frail individuals are lacking [218], which may be due to the exclusion of more vulnerable, older individuals from clinical trials [219]. Therefore, beyond vaccination, more measures for older individuals need to be implemented to decrease the burden of COVID-19 and improve outcomes related to the infection.

Further research and development prospects also concern prototype COVID-19 investigational vaccines administered via different delivery routes. Recent advances present the possibility of producing a peptide-directed phage particle that can be administered in an aerosolized form by inhalation. A combinatorial approach for ligand-directed pulmonary delivery as a unique route for systemic targeting in vaccination was shown to elicit a systemic and specific humoral response in mice and nonhuman primates [212]. In this approach, an immune epitope of SARS-CoV-2 S protein was engineered together with a small ligand peptide to ensure it crossed from the lungs into the systemic circulation, where it produced a strong antibody response in mice. The small peptide was a synthetic ligand capable of specifically binding to the host receptor expressed on the surface of the cells lining the lung airways and alveoli [220].

It is hoped that efforts towards the development of mucosal vaccines leading to secretory-IgA antibody production can provide a very strong first line of defense by preventing virus entry into the mucosa. It is becoming increasingly clear that local mucosal immune responses, which include, in addition to IgA antibodies, local mucosal IgG production and cytotoxic T lymphocyte activation, are very important for protection against COVID-19 disease. Despite current vaccines being capable of eliciting a strong systemic immune response by drastically boosting the development of neutralizing antibodies in serum, it seems that oral mucosal immunity is poorly activated [221]. The induction of front-line mucosal immunity offers the potential to mitigate the current and future respiratory virus epidemics and pandemics. In addition, maximizing individual protection against breakthrough infections can contribute to decreasing disease severity and the risk of virus transmission upon infection [222].

Therefore, mucosal COVID-19 vaccines represent a promise and a challenge, mainly due to needle-free administration and the ability to induce both mucosal (IgA) and circulating (IgG and IgA) antibodies as well as T-cell responses. Mucosal immune responses could also contribute to reducing the frequency of asymptomatic SARS-CoV-2-positive individuals, who represent an important factor in triggering and sustaining infection chains.

## 10. Conclusions

The most efficient strategy to combat the current COVID-19 pandemic and save millions of human lives worldwide is represented by active immunization. To date, approved vaccines have been shown to reduce both mortality and the incidence of severe COVID-19 and are today a fundamental weapon in the fight against SARS-CoV-2. However, the emergence of VOCs is still challenging vaccine-induced immune protection and underlines the need for multicoronavirus vaccine platforms capable of triggering a long-lasting protective immune response. In the meantime, current vaccination strategies, along with preventive measures, are the cornerstones of guaranteeing the best protection for all individuals.

## Figures and Tables

**Figure 1 vaccines-10-00608-f001:**
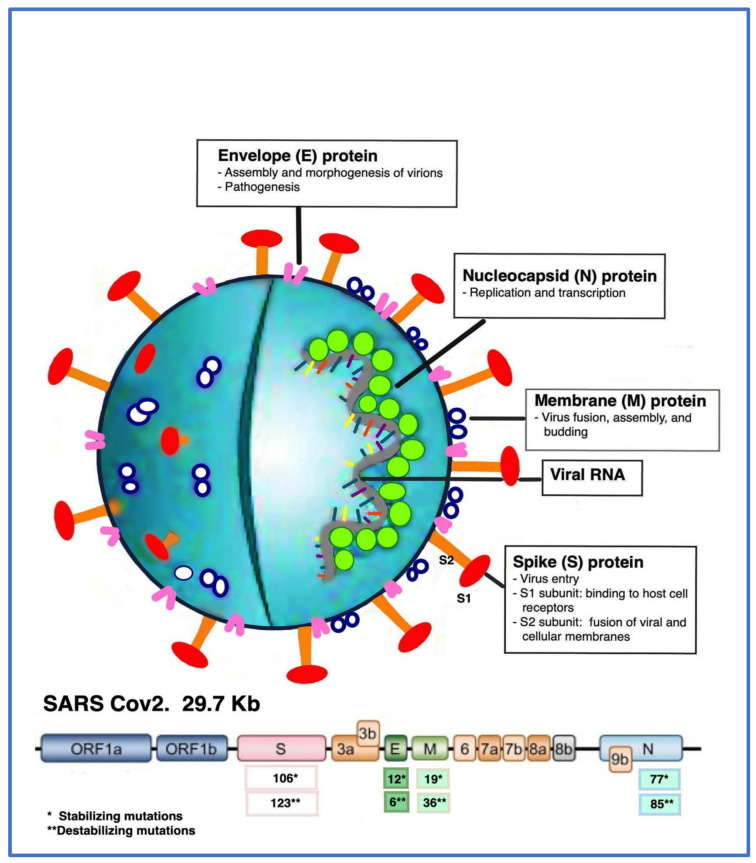
**Schematic representation of structure and genome of SARS-CoV-2.** The single-stranded RNA (ssRNA) of 29.7 Kb, with nucleocapsid (N) proteins, is covered with the envelope (E) and membrane (M) proteins, whereas the spike (S) proteins are located on the outside of the viral particle. The genome is also characterized by the presence of ORFs 1a and 1b, encoding nonstructural polyproteins (NSPs), and ORFs 3a, 3b, 6, 7a, 7b, 8a, 8b, and 9b, encoding accessory proteins. The main functions of structural proteins are also indicated as follows: E protein—virion assembly, morphogenesis, and pathogenesis; N protein—genome replication and transcription; M protein—virion fusion, assembly, and budding; S protein—virus entry, with S1 subunit binding to host cell receptors (RBDs) and S2 subunit fusing viral and cellular membranes. The numbers of stabilizing and destabilizing mutations identified in the different structural proteins of SARS-CoV-2 are also indicated.

**Figure 2 vaccines-10-00608-f002:**
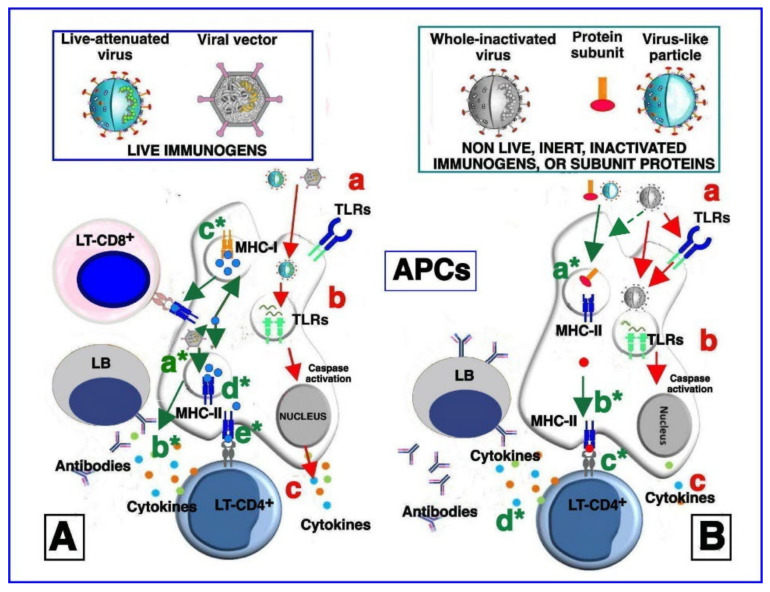
Simplified model of vaccine stimulation of the immune system. (**A**) **Immune stimulation by live-attenuated viruses and/or vectored viral immunogens.** After immunization, live-attenuated and vectored viral particles are endocytosed by antigen-presenting cells (APCs). The live-attenuated viruses can activate cell membrane pattern recognition receptors (PRRs) such as Toll-like receptors (TLRs) 2 and 6 (a). Upon entry, the live-attenuated viruses expose the nucleic acid and transcribe their genes, which, in turn, are sensed by endosomal TLRs (b). Activation of TLRs initiates signaling pathways culminating in the caspase pathway activation and production of pro-inflammatory and antiviral cytokines and chemokines (c). Immune stimulation by vectored viral immunogens may induce, via NOD-like receptor family pyrin domain-containing (NLRP) 3 pathway, inflammasome activation (a*) and cytokine production (b*). The transcribed vector-encoded transgene generates the immunogenic proteins (blue circles), which can then be proteosome-processed and associated with class I major histocompatibility complex (MHC-I) (c*) or with class II major histocompatibility complex (MHC-II) in endocytic vesicles (d*). MHC-I molecules loaded with transgenic epitopes translocate to the cell membrane, where they are recognized by antigen-specific CD8^+^ T cells (e*). Consequently, the infected cell is killed, and releases antigens in the extracellular space. In the same way, MHC-II molecules loaded with transgenic epitopes and translocated to the cell membrane are recognized by CD4^+^ helper T cells, which secrete cytokines and chemokines and further activate antigen-specific CD8^+^ T cells and B cells. Finally, stimulated B cells maturate into antibody-secreting plasma cells and/or memory B cells, as well as a portion of the stimulated T cells, which become memory cells (not shown). Overall, live immunogens are able to equally stimulate both humoral and cell-associated immune responses). (**B**) **Immune stimulation by non-live (inert) inactivated vaccines, protein subunits, and virus-like particles.** Immunization with antigens, inoculated together with the adjuvants that are added to vaccine formulations induces cytokine production from local cells. Cytokines in turn activate and/or attract APCs to the immunization site. In addition, the antigens may directly activate APCs through binding to cell membrane TLRs (a). The inactivated viruses are phagocytized by APCs and nucleic-acid traces inside the phagosomes and may activate endosomal TLRs (b), leading to the production of cytokines and chemokines (in a smaller amount, compared to (**A**) (**c**). The antigens contained in the inert vaccines in protein subunit formulations and in virus-like particles, after the entry in the call, are also degraded inside the endocytic vesicles, loaded onto MHC-II molecules (a*), and presented to CD4^+^ T cells (b*). Activation of CD4^+^ T lymphocytes leads to production of cytokines and chemokines, which induce the activation of antigen-specific B cells (c*), which maturate into antibody-secreting plasma cells (d*) and/or memory B cells. In general, inert antigens, such as nonlive inactivated vaccines, protein subunits, and virus-like particles, induce potent humoral responses and low-to-moderate T-cell responses. Activation of CD8^+^ T cells by inert antigens occurs through alternative pathways that are not depicted in this figure. The stimulation processes described and depicted in steps (a), (b), and (c) is reported less frequently and is less potent than stimulation by live immunogens (**A**), as depicted in (**B**) panel).

**Figure 3 vaccines-10-00608-f003:**
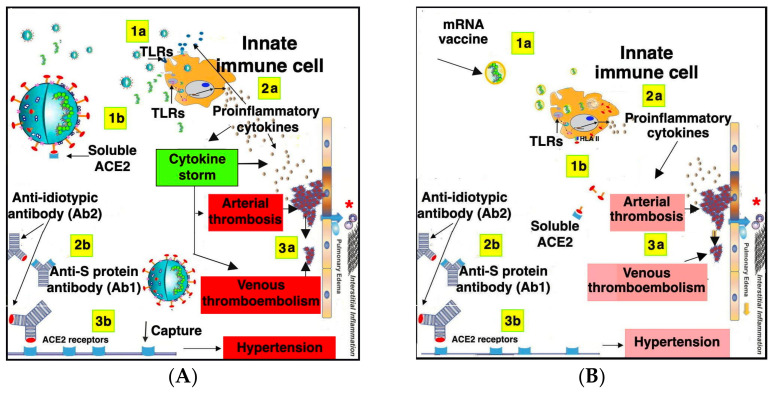
Comparative analysis of adverse reactions related to ACE2 following SARS-CoV-2 infection (**A**) and SARS-CoV-2 mRNA vaccination (**B**). (**A**) SARS-CoV-2 stimulates innate immune cells through both exogenous and endogenous TLR ligands (1a), inducing a strong TLR signaling that leads to cytokine storm (2a) which then induces a cascade of adverse events, including arterial thrombosis, venous thromboembolism, pulmonary edema, and interstitial inflammation ***** (3a). In addition, the S protein binds both to the ACE2 receptor and to soluble ACE2 (1b). S protein also induces Ab1 production (3b). Ab1, in turn, induces anti-idiotypic Ab2 production, which is able to bind to RBD of the anti-S protein antibodies as well as to both soluble ACE2 and ACE2 receptors, causing specific inhibition (3b). (**B**) Liposomal nanoparticles containing mRNA encoding viral S protein are phagocytized by innate immune cells (1a) and mRNA binds to specific endogenous TLRs. At the same time, mRNA molecules are translated into S proteins inside the immune cells that can then present them in association with HLA class II antigens, constitutively expressed on professional antigen-presenting cells, such as dendritic cells and monocytes/macrophages (1b). The binding of mRNA to endogenous TLRs induces more attenuated cytokine production (2a), which could lead to adverse events similar to those induced by SARS-CoV-2 infection, despite, in this case, being more attenuated and much less common (see the colored text boxes in lighter red, compared with red boxes in (**A**). The S protein presentation induces both humoral (2b) and cellular immune responses. Anti-idiotypic Ab2 production, with consequent inhibition of ACE2 activity, is also more attenuated compared with SARS-CoV2 infection).

**Table 1 vaccines-10-00608-t001:** Main nanoparticles for COVID-19 vaccines.

Vaccine Brand Name	Type of Nanoparticle	Technology	Role of Nanoparticles
mRNA-1273 ModernaTX, Inc., Cambridge (MA), United States	Lipid nanoparticles containing cholesterol, DSPC, and PEG2000	mRNA	RNA carrier for safe and efficient transport in vivo [148]
NVX-CoV2373	Virus-like nanoparticles containing saponin-based adjuvants	Protein subunit	Thermostable, higher binding affinity toward the human ACE2 receptor [149]
BNT162b2 Pfizer/BioNtech	Lipid nanoparticles containing an ionizable cationic lipid/phosphatidylcholine/cholesterol/PEG–lipid	mRNA	RNA carrier for safe and efficient transport in vivo [130,143,144]
ARCT-021 Arcturus Therapeutics Ltd.	Lipid nanoparticles containing ionizable lipid with a thioester to link the amine-bearing headgroup to lipid tails via two additional ester groups.	mRNA	RNA carrier for safe and efficient transport in vivo [150]
Chula Cov19 Chulalongkorn University	Novel lipid nanoparticles containing cationic lipids	mRNA	RNA carrier for safe and efficient transport in vivo [145]
CVnCoV CureVac AG	Lipid nanoparticles containing ionizable lipid ALC-0315	mRNA	RNA carrier for safe and efficient transport in vivo [146]

**Table 2 vaccines-10-00608-t002:** Main side effects of vaccines for COVID-19.

Vaccine Brand Name	Common Side Effects	Rare Side Effects
BNT162b2 Pfizer/BioNtech [201,202]	Pain at injection site, fatigue, headache, muscle and joint pain, chills, fever, diarrhoea	Nausea, vomiting, myocarditis, pericarditis, angioedema, anaphylaxis
SinoVac-CoronaVac [202,203]	Headache, fatigue, myalgia, nausea, abdominal pain, fever, diarrhea, pain at injection site	Vomit, ocular congestion, muscle spasms, hyposmia, nosebleed, constipation, hiccups
ChAdOx1-S, AZD1222 [202,204]	Chills, fever, joint or muscle pain, fatigue, headache, nausea, diarrhea, vomiting	Sleepiness, abdominal pain, lymphadenopathy, muscle spasms, thrombocytopenia, Guillain-Barrè syndrome, thrombosias, anaphylaxis
mRNA 1273 [202,205]	Fever, headache, fatigue, myalgia, arthralgia, nausea/vomiting, chills, pain at injection site, lymphadenopathy	Pericarditis, muscle spasms, paraesthesia/hyperaesthesia, sleep disturbances, cholelithiasis/cholecystitis, hypersensitivity
BBIBP CorV [179,202]	Flushing, swelling, fever, headache	Nasopharyngitis, drowsiness, palpitation

## Data Availability

Not applicable.

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
