# Peer review of "COVID-19 Vaccines: Current and Future Perspectives"

_vaccines, 2022, doi:10.3390/vaccines10040608_

Round 1

Reviewer 1 Report

This review is interesting and quite complete. However, I find that some part of it could be a little be more precise without being more complex. I only have minor comments: 

  • l73: This may be true for the Wuhan strain, but really less true for Delta or Omicron. Also, I don’t see the relevance of ref 16 here. Maybe you need to update your references to include more recent studies on Delta transmission?
  • l76: If you want to be complete you could also state other kind of non-respiratory symptoms, that you may find in more recent references than ref 17.
  • l178 and following: I tend to disagree with the term "natural infection" used here. It's clearly meant to differentiate with vaccination, but then "viral infection" would be enough, as you can "artificially infect" someone with a wild virus, so the process described hereafter is not specific to "natural infection". "Natural" is not specific enough and bears a wrong meaning in my opinion (as "natural" is often wrongly associated to something good).
  • l251: the list provided here should be consistent with what is covered below (order and topic).
  • l278: OPV has been used just everywhere in the world, where can’t you use live attenuated virus vaccines? The statement given here is wrong.
  • l288: Your reference is only on DNA vaccine, and what you say is obviously untrue regarding RNA vaccines against COVID19, unless anything below 90% efficacy is low immunogenicity.
  • l299: do you mean viral DNA or vaccine DNA ?
  • l553: You need to be more specific regarding the targeted virus and time of studies. The emergence of variants would change the results of the studies.
  • l574: The decrease is continuous not specifically after 6 months.
  • l610: ref 188 seems to be about thrombosis which is neither ADE or ERD. This part is a bit confusing, as it is said again below...
  • Writing style needs to be checked, potentially by a medical writer as some sentences are relatively heavy. 

Author Response

Reviewer 1

This review is interesting and quite complete. However, I find that some part of it could be a little be more precise without being more complex.

AU] We would like to thank the Reviewer for appreciating our work.

I only have minor comments: 

  • l73: This may be true for the Wuhan strain, but really less true for Delta or Omicron. Also, I don’t see the relevance of ref 16 here. Maybe you need to update your references to include more recent studies on Delta transmission?

AU] We completely agree with this comment. We updated references and reported studies regarding Delta and Omicron transmission (lines 71-78).

l76: If you want to be complete you could also state other kind of non-respiratory symptoms, that you may find in more recent references than ref 17. AU] We briefly reported description of non-respiratory symptoms of COVID-19 at lines 81-83.

  • l178 and following: I tend to disagree with the term "natural infection" used here. It's clearly meant to differentiate with vaccination, but then "viral infection" would be enough, as you can "artificially infect" someone with a wild virus, so the process described hereafter is not specific to "natural infection". "Natural" is not specific enough and bears a wrong meaning in my opinion (as "natural" is often wrongly associated to something good).

AU] Thank you for this comment. We completely agree with your suggestion and changed the term “natural infection” with “viral infection” throughout the manuscript.

  • l251: the list provided here should be consistent with what is covered below (order and topic).

AU] We checked and corrected the list (line 258-260).

  • l278: OPV has been used just everywhere in the world, where can’t you use live attenuated virus vaccines? The statement given here is wrong.

AU] We corrected the sentence accordingly (see lines 288-290).

  • l288: Your reference is only on DNA vaccine, and what you say is obviously untrue regarding RNA vaccines against COVID19, unless anything below 90% efficacy is low immunogenicity.

AU] We completely agree with you and corrected the sentence accordingly (see line 299).

  • l299: do you mean viral DNA or vaccine DNA ?

AU] We corrected the sentence accordingly, by changing viral with vaccine.

  • l553: You need to be more specific regarding the targeted virus and time of studies. The emergence of variants would change the results of the studies.

AU] We would like to thank the reviewer for this comment. We improved description of cited studies (see lines 564-575).

  • l574: The decrease is continuous not specifically after 6 months.

AU] We corrected the sentence accordingly (see line 585).

  • l610: ref 188 seems to be about thrombosis which is neither ADE or ERD. This part is a bit confusing, as it is said again below.... AU] We completely agree that this part was confusing; we then removed the incorrect reference and revised the writing style of this section.
  • Writing style needs to be checked, potentially by a medical writer as some sentences are relatively heavy. AU] Writing style of the manuscript was improved according to reviewer’s suggestions.

Reviewer 2 Report

The review article entitled ‘COVID-19 Vaccines: Current and Future Perspectives’ is very intriguing, well written, and embodies latest advancements in the field. Especially, the introduction section is very detailed and explains the modus operandi of the Coronaviruses and the latest state of the art Viral Vaccine platforms that are being used to counter the pandemic caused by the virus. There are a few suggestion related to the review article

  1. Please include a table with the latest developments of the clinical trails reported to the agencies, any adverse events of specific importance.
  2. A figure summarizing the various ways the Vaccines work in the Cell, would be more illustrative.

Author Response

The review article entitled ‘COVID-19 Vaccines: Current and Future Perspectives’ is very intriguing, well written, and embodies latest advancements in the field. Especially, the introduction section is very detailed and explains the modus operandi of the Coronaviruses and the latest state of the art Viral Vaccine platforms that are being used to counter the pandemic caused by the virus.

AU] We would like to thank the Reviewer for appreciating our work.

There are a few suggestion related to the review article

  1. Please include a table with the latest developments of the clinical trails reported to the agencies, any adverse events of specific importance.
  1. AU] We included a table with adverse events of vaccines reported in the latest clinical trials.
  1. A figure summarizing the various ways the Vaccines work in the Cell, would be more illustrative. We thank the reviewer for the suggestion; we add a third figure explaining mechanisms of action of distinct vaccine platforms on cell immune system.